# Colorectal Cancer Risk in Patients with Hemorrhoids: A 10-Year Population-Based Retrospective Cohort Study

**DOI:** 10.3390/ijerph18168655

**Published:** 2021-08-16

**Authors:** En-Bo Wu, Fung-Chang Sung, Cheng-Li Lin, Kuen-Lin Wu, Kuen-Bao Chen

**Affiliations:** 1Department of Anesthesiology, China Medical University Hospital, China Medical University, Taichung 404, Taiwan; enbofive@gmail.com; 2Department of Health Services Administration, College of Public Health, China Medical University, Taichung 404, Taiwan; fcsung1008@yahoo.com; 3Management Office for Health Data, China Medical University Hospital, Taichung 404, Taiwan; orangechengli@gmail.com; 4Department of Food Nutrition and Health Biotechnology, Asia University, Taichung 413, Taiwan; 5Division of Colorectal Surgery, Department of Surgery, Kaohsiung Chang-Gung Memorial Hospital and Chang-Gung University College of Medicine, Kaohsiung 833, Taiwan; kunn913@cgmh.org.tw; 6Department of Anesthesiology, College of Medicine, China Medical University, Taichung 404, Taiwan

**Keywords:** hemorrhoid, colorectal cancer, risk factor, propensity score matching, retrospective cohort study

## Abstract

Colorectal cancer (CRC) is a common disease and one of the leading causes of cancer deaths worldwide. This retrospective cohort study evaluated the risk of developing CRC in people with hemorrhoids. Using Taiwan’s National Health Insurance Research Database, we established three sets of retrospective study cohorts with and without hemorrhoids. The first set of cohorts were matched by sex and age, the second set of cohorts were matched by propensity score without including colonoscopies, and the third set of cohorts were matched by propensity score with colonoscopies, colorectal adenomas, and appendectomies included. In the second set of cohorts, 36,864 persons with hemorrhoids that were diagnosed from 2000 to 2010 and a comparison cohort, with the same size and matched by propensity score, were established and followed up to the end of 2011 to assess the incidence and Cox proportional regression-measured hazard ratio (HR) of CRC. The overall incidence rate of CRC was 2.39 times greater in the hemorrhoid cohort than it was in the comparison cohort (1.29 vs. 0.54 per 1000 person-years), with a multivariable model measured adjusted HR of 2.18 (95% CI = 1.78–2.67) after controlling for sex, age, and comorbidity. Further analysis on the CRC incidence rates among colorectal sites revealed higher incidence rates at the rectum and sigmoid than at other sites, with adjusted HRs 2.20 (95% CI = 1.48–3.28) and 1.79 (95% CI = 1.06–3.02), respectively. The overall incidence rates of both cohorts were similar in the first and second sets of cohorts, whereas the rate was lower in the third set of hemorrhoid cohorts than in the respective comparison cohorts, probably because of overmatching. Our findings suggest that patients with hemorrhoids were at an elevated risk of developing CRC. Colonoscopy may be strongly suggested for identifying CRC among those with hemorrhoids, especially if they have received a positive fecal occult blood test result.

## 1. Introduction

Colorectal cancer (CRC) is a common disease and one of the leading causes of cancer deaths worldwide. Both genetic and environmental factors are associated with the risk of developing CRC. The consumption of red meat, lack of physical activity, obesity, cigarette smoking, and alcohol use are known risk factors of CRC [1]. Most CRCs arise from adenomas that progress to dysplasia and carcinoma. On average, the progression from adenoma to adenocarcinoma takes 10 years [2]. Neoplastic changes result from both inherited and acquired genetic defects. Globally, CRC is the third most commonly diagnosed malignancy in males and second in females, and the incidences vary markedly in the World Health Organization GLOBOCAN database [3].

Hemorrhoids are common gastrointestinal disorders that are diagnosed by general practitioners. The disease is characterized by inflammation of the anorectum, including the submucosal, fibrovascular, and arteriovenous sinusoids [4]. Painless rectal bleeding is a common complaint during defecation, particularly when tissue prolapse appears. Other typical symptoms include anal pruritus, pain, and a lump at the anal verge due to thrombosis or strangulation [5]. A low-fiber diet and low water intake; increased intra-abdominal pressure caused by pregnancy, constipation, or prolonged straining; and weakened muscular support are risks factors for developing hemorrhoids [6,7]. The clinical manifestations of hemorrhoids are diverse, ranging from asymptomatic to rectal bleeding. Most people who have hemorrhoids are free of symptoms and usually do not require treatment [8].

CRC and hemorrhoids share several risk factors, such as low fiber intake, obesity, and lack of adequate exercise [9]. Hemorrhoids also present symptoms that are comparable to those of CRC, in particular, the presence of blood in the patient’s stool. Whether hemorrhoids are associated with the development of CRC is a known concern among the public, and consequently, screening for CRC among patients with hemorrhoids is often advisable as a precautionary measure. However, few studies have actually examined the relationship between CRC and hemorrhoids. Therefore, we conducted a population-based retrospective cohort study to explore the link between hemorrhoids and CRC using longitudinal insurance claims data from Taiwan.

## 2. Materials and Methods

### 2.1. Database

We obtained data from the National Health Research Institutes of Taiwan’s Longitudinal Health Insurance Database (LHID), which contains claims data of 1 million insured people that were randomly selected from 23 million registered users of the National Health Insurance (NHI) system for the period from 1996 to 2011. Information on patients’ demographics, health care received, and health care cost is available in the claims data. Diseases that were registered in the claims data can be identified using the International Classification of Diseases, Ninth Revision, Clinical Modification (ICD-9-CM). The identification numbers of those included in the data were scrambled with surrogate numbers to protect privacy. This study was approved by the Ethics Review Committee of China Medical University and Hospital (CMUH104-REC2-115-CR5).

### 2.2. Study Population and Comorbidity

From the LHID, we identified adult patients with hemorrhoids (ICD-9-CM 455) that were newly diagnosed from 2000 to 2010 as the hemorrhoid cohort. The first diagnosis date was considered as the index date. In order to delineate the role of a colonoscopy in CRC detection, we established three sets of the comparison cohorts without hemorrhoid histories from the LHID. In the first set, we randomly selected a comparison cohort with a sample size that was fourfold the size of the hemorrhoid cohort and was frequency matched by sex and age. In the second and third sets of cohorts, we established a comparison cohort with a sample size similar to the size of the hemorrhoid cohort, which was frequency matched by propensity score. The propensity score was estimated for each insured resident from a logistic regression with sociodemographic and potential comorbidities that are associated with cancer risk as covariates, including inflammatory bowel disease (IBD) (ICD-9-CM 555 and 556), hypertension (ICD-9-CM 401-405), diabetes (ICD-9-CM 250), hyperlipidemia (ICD-9-CM 272), stroke (ICD-9-CM 430-438), congestive heart failure (ICD-9-CM 428), obesity (ICD-9-CM 278), pyogenic liver abscess (PLA) (ICD-9-CM 572.0), hepatitis B virus (HBV) (ICD-9-CM V02.61, 070.20, 070.22, 070.30, and 070.32), hepatitis C virus (HCV) (ICD-9-CM V02.62, 070.41, 070.44, 070.51, 070.54, and 070.70-71), chronic obstructive pulmonary disease (COPD) (ICD-9-CM 491-492, and 496), alcohol-related illness (ICD-9-CM V11.3, A215, 291, 303, 305, 571.0-3, and 790.3) and chronic pancreatitis (ICD-9-CM 577.1). The histories of coloscopy use, colorectal adenomas, and appendectomy were also included in the propensity score estimation for the third set of study cohorts, but not for the second set of cohorts.

### 2.3. Statistical Analysis

We considered the second set of study cohorts to have had an appropriate design because the third set of study cohorts may have been overmatched by including colonoscopy, colorectal adenomas, and appendectomy as covariates in the estimation of the propensity score.

Data analyses first compared the characteristics of study subjects between the hemorrhoid and comparison cohorts. The distributions of categorical variables were examined using chi-squared testing and the means were examined using Student’s *t*-test. We estimated and plotted the cumulative incidence of CRC 1 year after the index date using the Kaplan–Meier method and tested the difference between curves using the log-rank test (Figure 1). The incidence density rates of CRC were also calculated 1 year after the index date. We used Cox proportional hazards regression analysis to calculate the hazard ratio (HR) and the corresponding 95% confidence interval (CI) of the hemorrhoid cohort against the comparison cohort. Multivariable analysis was used to estimate the adjusted hazard ratio (aHR). Overall incidence and incidence by colorectum site were also estimated, including in the ascending colon (ICD-9-CM 153.0 and 153.4-6), transverse colon (ICD-9-CM 153.1), descending colon (ICD-9-CM 153.2 and 153.7), sigmoid colon (ICD-9-CM 153.3), rectum (ICD-9-CM 154.0-1), and other locations (ICD-9-CM 153.8-9, 154.2-3, and 154.8). All statistical analysis was performed using SAS version 9.4 software (SAS Institute, Cary, NC, USA), and the figure of the cumulative incidence curve was plotted using R software. The significance level was set at *p* < 0.05 for two-sided testing.

Characteristics of the study population of the first and third sets of cohorts are presented in Appendix A. The overall incidence rates and HRs of CRC of the three study sets are presented in Appendix A.

## 3. Results

In this section, we present the findings that were based on the data for the second set of study cohorts, which had equal sample sizes of 36,864 persons.

### 3.1. Demography

These propensity-score-matched study cohorts were similar in prevalence rates of most comorbidities, except that the rates of hypertension, diabetes, and congestive heart failure were higher in the hemorrhoid cohort than in the comparison cohort (Table 1). More than half of the study population were men and there were more in the hemorrhoid cohort than there were in the comparison cohort. The hemorrhoid cohort was older than the comparison cohort was.

### 3.2. Higher Incidence Rates of CRC in Hemorrhoids Patients

Figure 1 shows that the cumulative incidence of colon cancer was 0.94% greater in the hemorrhoid cohort than it was in the comparison cohort (log-rank test *p* < 0.001).

### 3.3. Subgroup Analysis of Demographic Aspects and Comorbidities

After mean follow-up durations of 6.99 (SD = 3.12) and 6.85 years (SD = 3.13), the incidence rate of CRC was 2.39-fold greater in patients with hemorrhoids than it was in the comparison cohort, respectively (Table 2). After adjusting for age and gender, the hemorrhoids cohort had an adjusted HR of 2.18 (95% CI = 1.78–2.67) for CRC. The CRC incidence increased with age and was higher in men than it was in women in both cohorts, whereas the adjusted HR of CRC was slightly higher for women than it was for men. Relative to the comparison group, the HRs of colon cancer were 3.53 (95% CI = 2.15–5.79), 1.91 (95% CI = 1.36–2.68), and 2.06 (95% CI = 1.53–2.77) for patients with hemorrhoids aged ≤49, 50–64, and ≥65 years, respectively. Comorbidity was also associated with an excess increase in CRC incidence, which was greater in the hemorrhoid cohort than in the comparison cohort (1.49 versus 0.56 per 1000 person-years) compared to those without comorbidity.

### 3.4. Relevance of Tumor Site and Hemorrhoids

Table 3 presents the occurrence of CRC by site for both cohorts. The cancer incidence was greater in the hemorrhoid cohort than it was in the comparison cohort for most sites, particularly for cancer of the rectum and sigmoid colon, with adjusted HRs of 2.20 (95% CI = 1.48–3.28) and 1.79 (95% CI = 1.06–3.02), respectively.

## 4. Discussion

This study established three sets of study cohorts. The sex- and age-matched hemorrhoid cohort had more prevalent baseline comorbidities than the comparison cohort (Appendix A). The second and third sets of cohorts were propensity score matched, which had similar distributions between hemorrhoid and comparison cohorts for most comorbidities. We found that the overall CRC incidence rates were similar in the first and second sets of cohorts, with 1.29 per 1000 person-years in the hemorrhoids cohorts and 0.54 per 1000 person-years in the comparison cohorts (Table 2 and Appendix A). In the third set of study cohorts, the CRC incidence was lower in the hemorrhoids cohort than it was in the comparison cohort (1.22 versus 1.62 per 1000 person-years), with an aHR of 0.80 (95% CI = 0.69, 0.94). We suspected that the third set of study cohorts were overmatched by including colonoscopies, colorectal adenomas, and appendectomies in the propensity score matching, in which both cohorts had similar prevalence rates of colonoscopy uses at baseline. The prevalence of colorectal adenomas was even higher in the hemorrhoid cohort than in the comparison cohort. Therefore, we considered the second set of study cohorts as being adequate for studying the relationship between hemorrhoids and CRC risk.

The advantage of this study was using a large data set to conduct a nationwide population study with a long follow-up period [10]. The follow-up was started 12 months after the index date to avoid the immortal effect. Therefore, the association between hemorrhoids and the risk of CRC demonstrated in the present study was highly convincing [11].

CRC is a common malignancy that results in more than 600,000 deaths globally every year [12]. Among the risk factors that are associated with this cancer, genetic factors account for approximately 20% of all cases of the disease [13,14]. Other instances can be associated with environmental causes rather than genetic factors. The typical pathophysiology of CRC development has a time frame of a decade, with the polyp-to-adenocarcinoma stage averaging 7–10 years [15]. Dysplastic adenomas are the most common form of premalignant lesions [16]. Genetic features show that adenomatous polyposis coli gene mutations are present in the early phase of CRC formation in approximately 70% of adenomas, with 49–50% caused by *KRAS* mutations [17]. Subsequently, activated *KRAS* oncogene mutations and inactivated mutations of the *TP53* tumor suppressor gene can result in an adenoma–carcinoma sequence. Comprehensive genome analyses have found *APC*, *KRAS*, *TGFBR2*, *SMAD4*, and *TP53* as driver genes with mutations frequently found in human colorectal cancers [18,19].

To our knowledge, environmental factors and genetic factors can promote the formation of CRC. Among various environmental factors, inflammation was implicated in elevated cancer risk [20,21,22]. Studies have indicated the vital roles of IBD, Crohn’s disease, and ulcerative colitis in the development of CRC [23,24,25]. Studies also discovered that upregulated interleukin-17, interleukin-23, and signal transducer and activator of transcription results in tumor development in patients with intestinal inflammation [26,27].

However, a recent study has recommended low-dose aspirin for the chemoprevention of CRC [28]. Instances in which the benefits outweigh the risks have not yet been clearly defined for specific individuals. A prospective, double-blind, multidose, and placebo-controlled clinical trial, namely, ASPIRED, proposed that acetylsalicylic acid may be related to the inhibition of cyclooxygenase’s inflammatory mechanism and other inflammatory biomarkers in the formation of CRC [29].

In an earlier case–control study in the United States, Tseng et al. reported that hemorrhoids, sexually transmitted diseases, physical inactivity, and multiple sexual partners are factors that are associated with an increased risk of anal cancer [30]. Frequent receptive anal intercourse is likely the physio-pathological cause that leads to inflammation and cancer development. Information on sexual activity and physical inactivity is not available in our data to evaluate these relationships. Another case–control study in San Francisco also found the homosexual activity and hemorrhoids are associated with the development of anal and rectal squamous cell carcinoma [31]. Long-lasting hemorrhoids may act similarly to chronic inflammatory dermatoses, which may result in regional inflammatory reactions and evolve in the localized squamous-cell carcinomas. In our study, we also found that hemorrhoids significantly increase the risk of distal colon cancer, including in the sigmoid colon, rectum, anus, and other sites (Table 3). This finding implicates that localized inflammation may lead to tumor formation. In addition, epigenetic factors may also be associated with CRC risk. In a Swedish case–control study, Ungerbäck et al. examined polymorphisms in nuclear factor kappa B (NF-*κ*B)-pathway-associated genes, which are the major regulators of inflammation [32]. The results showed that advanced CRC is associated with heterozygous and polymorphic *TNFAIP3 (rs6920220)*, heterozygous *NLRP3 (Q705K)*, and polymorphic *NF-**κB-94 ATTG ins/del* genotypes. The proinflammatory NF-*κ*B pathway degenerates the tissue, increases the risk of sustained cell and DNA damage, and promotes other tumor biomarkers that are activated in the primary and metastatic CRC. These studies demonstrated that chronic inflammation is associated with the formation of CRC.

To date, few large-scale studies have investigated the relationship between hemorrhoids and CRC. In our retrospective cohort study with a 12-year follow-up period, we found that patients with hemorrhoids had an aHR of 2.18 for developing CRC after matching comorbidities, some of which are proven risk factors for CRC [33], and adjusting for age and gender. In a population-based retrospective cohort study in 2013, Lee et al. observed that the presence of hemorrhoids was significantly associated with a vital risk of developing CRC [34], as shown by an increased standardized incidence ratio (SIR) of 1.50. The SIR method has a limitation, namely, that the effects of comorbidities cannot be analyzed. We contend that adjustment for confounding factors is crucial in big data studies. The propensity score matching method creates a balanced dataset, allowing for a simple and direct comparison of baseline covariates between experimental and control groups. After matching the propensity score, it is used to derive a pseudo-randomized dataset, which allows for unbiased estimation of the exposure effect. Therefore, matching based on the propensity score is frequently used in the analysis of medical statistics. Our study provides stronger evidence of a link between hemorrhoids and CRC after propensity score matching and adjusting for associated risks. Our study also has the advantage of estimating the risk variation between different colorectal sites. These results imply that the location of colorectal cancer is consistent with our conclusions. However, whether the shorter latent period for CRC with hemorrhoids is attributable to molecular changes is unclear. Genetic or environmental dispositions may exist in Taiwanese patients and this deserves further investigation.

## 5. Advantages and Limitations

This study was strengthened by using a longitudinal design with a high follow-up rate and was based on population-based data with a reliable diagnosis. Taiwan’s LHID was created to advance the health of all citizens. Health representatives from other countries were interested in visiting with the Taiwan National Health Insurance Administration (NHIA) to learn about the insurance program. It has been reported that 981 dignitaries from 71 countries had visited the agency by the end of 2016 [10].

However, there are several limitations to our study. First, lifestyle information regarding drinking, smoking, and diet, as well as genetic factors, were unavailable to observe the related impacts. Second, the propensity score matching method was used to establish study cohorts; the bias resulting from the retrospective nature of the study may have influenced our findings. Third, all data involved were anonymous and patients were selected through ICD codes; therefore, the accuracy of the diagnoses can be disputed. Fourth, patients with hemorrhoids might be more likely to be screened for cancer and were more likely to be diagnosed with the disease. We conducted further data analysis to compare the history of colonoscopy examinations between cohorts with and without hemorrhoids. Results showed that the hemorrhoid group had colonoscopies more frequently than the comparison cohort had, particularly during the follow-up period (Appendix A). However, we found that hemorrhoid surgery could reduce the colorectal cancer incidence by 49.3%, from 1.46 to 0.74 per 1000 person-years (Appendix A). We also found that the hemorrhoid cohort without a hemorrhage of the rectum and/or anus (ICD-9-CM 569.3) were at higher risk for colorectal cancer with an aHR of 2.09 (Appendix A). These data may imply that hemorrhoids could be associated with a twofold increased risk without the surgery. Therefore, the claim that hemorrhoids are associated with an increased risk of CRC remains reliable.

## 6. Conclusions

Our propensity-score-matched retrospective study demonstrated that patients with hemorrhoids were at a nearly twofold increased risk of developing CRC. Hemorrhoidectomy is seen as a benefit to hemorrhoid patients since it provides a near 50% risk reduction of subsequent CRC. CRC is one of the most critical findings of colonoscopies, which is used for polypectomy and biopsy if any suspicious lesions are found. On the basis of these findings, we propose that hemorrhoids are a specific risk factor for CRC. Consequently, people with hemorrhoids should be encouraged to undergo a colonoscopy for the early detection of CRC.

## Figures and Tables

**Figure 1 ijerph-18-08655-f001:**
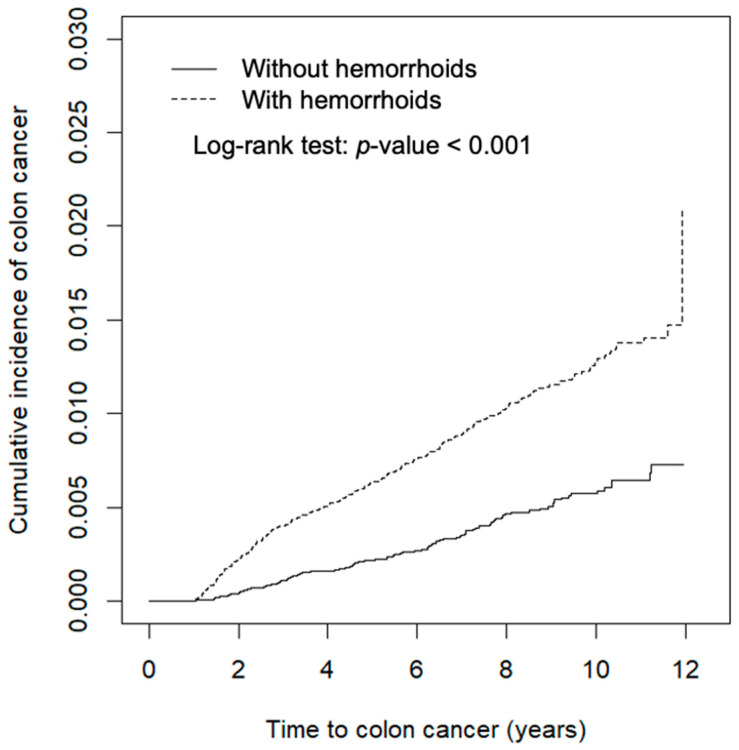
Cumulative incidence of colorectal cancer of cohorts with and without hemorrhoids, as obtained using the Kaplan–Meier method.

**Table 1 ijerph-18-08655-t001:** Demographic characteristics and comorbidities that were compared between cohorts with and without hemorrhoids who were matched by propensity score without colonoscopies.

Variable	Hemorrhoids		Standardized Difference ^§^
	No	Yes	
	*n* = 36,864	*n* = 36,864	
Sex	*n* (%)	*n* (%)	
Female	17,582 (47.7)	16,809 (45.6)	0.04
Male	19,282 (52.3)	20,055 (54.4)	0.04
Age, mean (SD)	45.8 (15.9)	46.9 (15.8)	0.07
Stratified			
≤49 years	23,569 (63.9)	22,622 (61.4)	0.05
50–64	7836 (21.3)	8443 (22.9)	0.04
65+	5459 (14.8)	5799 (15.7)	0.03
Comorbidity			
IBD	566 (1.54)	585 (1.59)	0.004
Hypertension	8580 (23.3)	9694 (26.3)	0.07
Diabetes	2120 (5.75)	2252 (6.11)	0.02
Hyperlipidemia	7206 (19.6)	7263 (19.7)	0.004
Stroke	830 (2.25)	894 (2.43)	0.01
Congestive heart failure	984 (2.67)	1116 (3.03)	0.02
Obesity	173 (0.47)	193 (0.52)	0.008
PLA	22 (0.06)	23 (0.06)	0.001
HBV	1512 (4.10)	1512 (4.10)	0.000
HCV	452 (1.23)	505 (1.37)	0.01
COPD	3879 (10.5)	3860 (10.5)	0.002
Alcohol-related illness	1395 (3.78)	1397 (3.79)	0.000
Chronic pancreatitis	46 (0.12)	62 (0.17)	0.01

^§^ A standardized mean difference of ≤0.10 indicates a negligible difference between the two cohorts. SD, standard deviation; IBD, inflammatory bowel disease; PLA, pyogenic liver abscess; HBV, hepatitis B virus; HCV, hepatitis C virus; COPD, chronic obstructive pulmonary disease.

**Table 2 ijerph-18-08655-t002:** Incidence and hazard ratio of colorectal cancer compared between the cohorts with and without hemorrhoids by gender, age, and comorbidity.

Variable	Hemorrhoids						Crude HR (95% CI)	Adjusted HR (95% CI)
	No			Yes				
	Event	PY	Rate ^#^	Event	PY	Rate ^#^		
All	138	255,722	0.54	337	261,466	1.29	2.39 (1.96, 2.93) *	2.18 (1.78, 2.67) *
Gender								
Female	52	113,232	0.46	128	121,173	1.06	2.35 (1.71, 3.25) *	2.19 (1.59, 3.03) *
Male	86	142,490	0.60	209	140,293	1.49	2.38 (1.84, 3.09) *	2.16 (1.66, 2.80) *
Age-stratified								
≤49	23	162,628	0.14	71	165,694	0.43	3.53 (2.15, 5.79) *	3.53 (2.15, 5.79) *
50–64	45	55,159	0.82	113	57,499	1.97	1.95 (1.39, 2.73) *	1.91 (1.36, 2.68) *
65+	70	37,935	1.85	153	38,272	4.00	2.12 (1.58, 2.85) *	2.06 (1.53, 2.77) *
Comorbidity								
No	54	161,577	0.33	98	149,717	0.65	2.58 (1.79, 3.72) *	2.53 (1.76, 3.66) *
Yes	84	94,145	0.89	239	111,749	2.14	2.13 (1.67, 2.72) *	2.09 (1.64, 2.67) *

Rate ^#^: incidence rate per 1000 person-years; PY, person-years; HR, hazard ratio. Adjusted HR: estimated after controlling for sex, age, and comorbidity. Comorbidity: patients with any one of the comorbidities (IBD, hypertension, diabetes, hyperlipidemia, stroke, congestive heart failure, obesity, PLA, HBV, HCV, COPD, alcohol-related illness, and chronic pancreatitis) were assigned to the comorbidity group. * *p* < 0.001.

**Table 3 ijerph-18-08655-t003:** Incidences and hazard ratio of colorectal cancer estimated by colorectum site compared between cohorts with and without hemorrhoids.

Site of CRC	Hemorrhoids				Crude HR (95% CI)	Adjusted HR (95% CI)
	No		Yes			
	Event	Rate ^#^	Event	Rate ^#^		
Ascending colon (ICD-9-CM 153.0, 153.4-6)	13	0.05	21	0.08	1.49 (0.75, 2.97)	1.34 (0.67, 2.69)
Transverse colon (ICD-9-CM 153.1)	4	0.02	5	0.02	1.18 (0.32, 4.41)	1.08 (0.29, 4.07)
Descending colon (ICD-9-CM 153.2, 153.7)	3	0.01	10	0.04	3.13 (0.86, 11.4)	2.83 (0.78, 10.3)
Sigmoid colon (ICD-9-CM 153.3)	21	0.08	43	0.17	1.92 (1.14, 3.24) *	1.79 (1.06, 3.02) *
Rectum (ICD-9-CM 154.0-1)	34	0.14	86	0.33	2.40 (1.61, 3.57) **	2.20 (1.48, 3.28) **
Other (ICD-9-CM 153.8-9, 154.2-3, 154.8)	18	0.07	39	0.15	2.08 (1.19, 3.64)	1.86 (1.06, 3.26)
Unknown	45	0.17	133	0.51	2.99 (1.97, 3.66) **	2.76 (1.85, 3.67) **

Rate ^#^: incidence rate per 1000 person-years; HR, hazard ratio. Adjusted HR: estimated after controlling for sex, age, and comorbidity. * *p* < 0.05, ** *p* < 0.001.

## Data Availability

Data are available from the NHIRD published by Taiwan NHIA. Due to legal restrictions imposed by the government of Taiwan in relation to the “Personal Information Protection Act,” the data cannot be made publicly available. Requests for data can be sent as a formal proposal to the NHIRD (http://nhird.nhri.org.tw, accessed on 30 June 2021).

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
