# Peer review of "Colorectal Cancer Risk in Patients with Hemorrhoids: A 10-Year Population-Based Retrospective Cohort Study"

_ijerph, 2021, doi:10.3390/ijerph18168655_

Round 1

Reviewer 1 Report

The presence of hemorrhoids appears as a risk factor, but the right reasons cannot be deduced from the article. Instead, it should be emphasized that in patients with hemorrhoids, unfortunately, the diagnosis of a possible polyp is delayed, mistaking the symptoms of blood loss for simple hemorrhoids. Meanwhile, the polyp turns into cancer (as most CRCs start with a benign polyp that stays that way for 5-10 years, before turning into a CRC).

In patients who do not have hemorrhoids, however, the presence of rectal hemorrhages immediately makes both the patient and his doctor suspicious and this leads them to request a colonoscopy that can then detect the polyp and, consequently, stop its evolution towards neoplasm ( by perendoscopic polypectomy). That's why, in my opinion, people who don't have hemorrhoids are less likely to develop colorectal cancer: blood loss isn't confused with hemorrhoidal bleeding.
The authors on p. line, they write but do not indicate the bibliography and I have not found any. The statistical methodology has been applied correctly, but unfortunately in this case the scientific prerequisites to consider the results valid are lacking. Add to this, an important disadvantage of the Cox proportional hazards regression analysis method, when used in the presence of confounding variables, as in this case, because this method does not consider the joint distributions of the confounding variables.

An alternative way to address confounding in cohort studies is the use of propensity scores, a method developed by Rosenbaum & Rubin which can reduce bias in these studies. Or you can use it the multivariate confounding score suggested by Miettinen, very similar to the propensity score. In light of all this, it is advisable to review the collected data and read them under a different interpretation, and to adopt more appropriate and less biased statistical techniques.

Author Response

#Response to Reviewer 1 Comments

Manuscript ID: ijerph-1244969
2nd Revision for- Colorectal Cancer Risk in Patients with Hemorrhoids: A 10-year Population-Based Retrospective Cohort Study

Point 1: It should be emphasized that in patients with hemorrhoids, unfortunately, the diagnosis of a possible polyp is delayed, mistaking the symptoms of blood loss for simple hemorrhoids.

Response 1: Thank you for the comment. We appreciate your inspirational points of view.

More than 98% of residents in Taiwan have been covered in the insurance program. Insured people are encouraged to have occult blood examination, and the 40s years old people with family history or 50+ years old ones are encouraged to have colonoscopy. (Reference: Health Promotion Administration, Ministry of Health and Welfare of Taiwan https://www.hpa.gov.tw/EngPages/Detail.aspx?nodeid=1038&pid=7615)

We have followed Reviewers’ comment # 4 to establish propensity score matched cohorts with and without hemorrhoid. Based on these cohorts, we further analyzed the data, results (Supplement Table 3) showed that the hemorrhoids cohort without hemorrhage of rectum and anus were at higher risk for colorectal cancer with an adjusted HR of 2.09. On the other hand, for those with hemorrhage of rectum and anus, there was no risk difference between the 2 cohorts. Similar relationship appeared for those with and without hemorrhage of gastrointestinal tract, unspecified. Clinicians usually encourage patients with bleeding to the colonoscopy, which may increase polypectomy. Polypectomy prevents benign polyps (adenoma) developing malignant (adenocarcinoma).

     Supplement Table 3. Incidence and hazard ratios of colorectal cancer associated with polyp or hemorrhage

 compared between the cohorts with and without hemorrhoids.

Hemorrhoids

No

Yes

Variable

Event

PY

Rate#

Event

PY

Rate#

Crude HR

Adjusted HR†          (95% CI)

(95% CI)

Polyp or hemorrhage of rectum and anus

No

124

245766

0.50

266

228979

1.16

2.29 (1.85, 2.84)***

2.09 (1.69, 2.59)***

Yes

7

1657

4.22

64

29914

2.14

0.50 (0.23, 1.10)

0.59 (0.27, 1.31)

Hemorrhage of gastrointestinal tract, unspecified

No

102

234739

0.43

226

219621

1.03

2.35 (1.86, 2.97)***

2.03 (1.60, 2.59)***

Yes

29

12684

2.29

104

39273

2.65

1.16 (0.77, 1.74)

1.27 (0.83, 1.94)

               Rate#, incidence rate per 1000 person-years; PY person-years, HR hazard ratio. Adjusted HR: adjusted for age,

               sex and comorbidity. *P < 0.05, **P < 0.01, ***P < 0.001.

Point 2: In patients who do not have hemorrhoids, however, the presence of rectal hemorrhages immediately makes both the patient and his doctor suspicious and this leads them to request a colonoscopy that can then detect the polyp and, consequently, stop its evolution towards neoplasm (by panendoscopic polypectomy). That's why, in my opinion, people who don't have hemorrhoids are less likely to develop colorectal cancer: blood loss isn't confused with hemorrhoidal bleeding.

Response 2: Thank you for the comment. We appreciate your points of view.  

We agree that a sudden low GI bleeding in patients without hemorrhoids would also alert clinicians for a colonoscopy. For the comment # 1, we have replied that polypectomy is likely to be ordered for patients with bleeding. We appreciate your points of view, which makes more patients receive polypectomy. Polypectomy prevents benign polyps (adenoma) developing malignant (adenocarcinoma).

Yes, frequent colonoscopy procedure would increase colorectal cancer to be diagnosed.

For this reason, we have conducted another data analysis based on the propensity score matched cohorts. Results (Supplement Table 1) show that patients with hemorrhoids cohort are more likely to receive colonoscopy at the baseline and during the follow-up period, and the diagnosis of the cancer is thus increased. In the revision, we have replied in the study Limitation:

“Fourth, patients with hemorrhoids might be more likely to be screened for cancer and were more likely to be diagnosed with the disease. We have conducted further data analysis to compare the history of colonoscopy examinations between cohorts with without hemorrhoid. Results showed that the hemorrhoid group had used colonoscopy more frequently than comparisons had, particularly during the follow-up period (Supplement Table 1). However, we found that hemorrhoid surgery could reduce 49.3% of colorectal cancer incidence, from 1.46 to 0.74 per 1000 person-years (Supplement Table 2). We also found that the hemorrhoids cohort without hemorrhage of rectum and anus were at higher risk for colorectal cancer with an adjusted HR of 2.09 (Supplement Table 3). These data may imply that hemorrhoid could be associated with a 2-fold increased risk without the surgery. Therefore, hemorrhoids are associated with increased risk of CRC remain reliable.” (Please see lines 251-261.)

                               Supplement Table 1. Number of colonoscopy received before and after baseline

                        compared between cohorts with and without hemorrhoid.

Hemorrhoids

Variable

No

N=36864

Yes

N=36864

Colonoscopy

n

Rate

n

Rate

p-value

Before

<0.001

0

35993

97.6

32110

87.1

1

732

1.99

3926

10.7

2

102

0.28

614

1.67

≧3

37

0.10

214

0.58

After

<0.001

0

35115

95.3

26338

71.5

1

1391

3.77

6994

19.0

2

253

0.69

2193

5.95

≧3

105

0.28

1339

3.63

Supplement Table 2. Incidence and hazard ratio of colorectal cancer associated with surgery of hemorrhoids

Variables

N

Event

Rate#

Crude

HR (95% CI)

Adjusted

HR (95% CI) 

  Non-hemorrhoids

36864

131

0.53

1.00

1.00

Hemorrhoids

Surgery

No

27783

280

1.46

2.76(2.24, 3.39)***

2.31(1.88, 2.85)***

Yes

9081

50

0.74

1.38(0.99, 1.91)

1.65(1.19, 2.29)**

                    Rate#, incidence per 1000 person-years; HR, hazard ratio. Adjusted HR: controlling for sex and age.

                    *P < 0.05, **P < 0.01, ***P < 0.001.

Point 3: The authors on p. line, they write but do not indicate the bibliography and I have not found any.

Response 3: Thank you for the comment. We apology for the mistake. We have cited the reference in the revision please see in Line 170.

“Among the risk factors associated with this cancer, genetic factors account for approximately 20% of all cases of the disease [13,14].”

(Rustgi, A.K. The genetics of hereditary colon cancer. Genes Dev 2007, 21, 2525-2538, doi:10.1101/gad.1593107.

Tenesa, A.; Dunlop, M.G. New insights into the aetiology of colorectal cancer from genome-wide association studies. Nat Rev Genet 2009, 10, 353-358, doi:10.1038/nrg2574.)

Point 4: An important disadvantage of the Cox proportional hazards regression analysis method, when used in the presence of confounding variables, as in this case, because this method does not consider the joint distributions of the confounding variables. An alternative way to address confounding in cohort studies is the use of propensity scores, a method developed by Rosenbaum & Rubin which can reduce bias in these studies. Or you can use it the multivariate confounding score suggested by Miettinen, very similar to the propensity score.

Response 4: Thank you for the comment and inspirational suggestion.

For the revision, we have followed your suggestion and re-designed the study by establishing propensity score matched cohorts with and without hemorrhoid. Most prevalence rates of baseline comorbidities were similar between the 2 cohorts. Results show that the risk of developing colorectal cancer remains higher in the hemorrhoid cohort than in comparisons.   

Table 2. Incidence and hazard ratio of colon cancer compared between the cohorts with and without hemorrhoids by gender, age, and comorbidity.

Hemorrhoids

No

Yes

Variable

Event

PY

Rate#

Event

PY

Rate#

Crude HR

Adjusted HR†          (95% CI)

(95% CI)

All

138

255722

0.54

337

261466

1.29

2.39(1.96, 2.93)***

2.18(1.78, 2.67)***

Gender

Female

52

113232

0.46

128

121173

1.06

2.35(1.71, 3.25)***

2.19(1.59, 3.03)***

Male

86

142490

0.60

209

140293

1.49

2.38(1.84, 3.09)***

2.16(1.66, 2.80)***

Stratify age

≤49

23

162628

0.14

71

165694

0.43

3.53(2.15, 5.79)***

3.53(2.15, 5.79)***

50-64

45

55159

0.82

113

57499

1.97

1.95(1.39, 2.73)***

1.91(1.36, 2.68)***

65+

70

37935

1.85

153

38272

4.00

2.12(1.58, 2.85)***

2.06(1.53, 2.77)***

Comorbidity‡

No

54

161577

0.33

98

149717

0.65

2.58(1.79, 3.72)***

2.53(1.76, 3.66)***

Yes

84

94145

0.89

239

111749

2.14

2.13(1.67, 2.72)***

2.09(1.64, 2.67)***

              Rate#, incidence rate per 1000 person-years; PY person-years, HR hazard ratio.

             Adjusted HR: multivariable analysis of age, and gender.

             Comorbidity‡: patients with any one of the comorbidities (IBD, hypertension, diabetes, hyperlipidemia, stroke,            

             congestive heart failure, obesity, PLA, HBV, HCV, COPD, alcohol-related illness, and chronic pancreatitis) were

             classified as the comorbidity group. *P < 0.05, **P < 0.01, ***P < 0.001.

Reviewer 2 Report

Thank you for giving me a chance to review this manuscript. I generally do not disagree with the content. However, it is possible that an important confounding bias has been overlooked in this study.

If most cases of hemorrhoids were diagnosed by endoscopy, then more people who underwent colonoscopy would be included in the group with hemorrhoids than in the group without hemorrhoids. In addition, the majority of patients with CRC have undergone colonoscopy during the course of diagnosis and treatment. As a result, those who have undergone colonoscopy have a higher prevalence of CRC than those who have not undergone colonoscopy, and at the same time, more hemorrhoids are diagnosed.

Although there may be limitations in the information that can be gleaned from the database of National Health Insurance, this study may be influenced by an important confounding factor: the presence or absence of colonoscopy. I think that the authors should re-examine this analysis considering history of colonoscopy. 

Author Response

#Response to Reviewer 2 Comments

Manuscript ID: ijerph-1244969
2nd Revision for- Colorectal Cancer Risk in Patients with Hemorrhoids: A 10-year Population-Based Retrospective Cohort Study

Point 1: If most cases of hemorrhoids were diagnosed by endoscopy, then more people who underwent colonoscopy would be included in the group with hemorrhoids than in the group without hemorrhoids.

Response 1: Thank you for the comment.

In general, the diagnosis of hemorrhoids in Taiwan is not initiated with colonoscopy. Patients with hemorrhoid complaints are usually diagnosed by clinical inspections, digital examinations, and anoscopes in outpatient visits. Physicians order interventions with medicine or surgery to patients based on the severity of symptoms.

It is true people who underwent colonoscopy would be more likely included in the group with hemorrhoids than in the group without hemorrhoids. We have re-designed our study for cohorts with and without hemorrhoid by establishing propensity score matched cohorts. Based on these cohorts, we further analyzed the data considering the history of colonoscopy. Please see our findings in the reply to comment point 2, next page.  

Point 2: In addition, the majority of patients with colorectal cancer have undergone

colonoscopy during the course of diagnosis and treatment. As a result, those who have undergone colonoscopy have a higher prevalence of colorectal cancer than those who have not undergone colonoscopy, and at the same time, more hemorrhoids are diagnosed. Although there may be limitations in the information that can be gleaned from the database of National Health Insurance, this study may be influenced by an important confounding factor: the presence or absence of colonoscopy. I think that the authors should re-examine this analysis considering history of colonoscopy.

Response 2: Thank you for the inspirational comment and suggestion.

Yes, the presence or absence of colonoscopy is an important factor associated with the diagnosis of colorectal cancer. In the re-designed and established cohorts with and without hemorrhoid with propensity score matching, we conducted an additional data analysis to compare the history of colonoscopy. Results show that the hemorrhoid cohort received the colonoscopy exams more often than the comparisons had, particularly during the follow-up period. In the revision, we have replied in the study Limitation:

“Fourth, patients with hemorrhoids might be more likely to be screened for cancer and were more likely to be diagnosed with the disease. We have conducted further data analysis to compare the history of colonoscopy examinations between cohorts with without hemorrhoid. Results showed that the hemorrhoid group had used colonoscopy more frequently than comparisons had, particularly during the follow-up period (Supplement Table 1). However, we found that hemorrhoid surgery could reduce 49.3% of colorectal cancer incidence, from 1.46 to 0.74 per 1000 person-years (Supplement Table 2). We also found that the hemorrhoids cohort without hemorrhage of rectum and anus were at higher risk for colorectal cancer with an aHR of 2.09 (Supplement Table 3). These data may imply that hemorrhoid could be associated with a 2-fold increased risk without the surgery. Therefore, hemorrhoids are associated with increased risk of CRC remain reliable.” (Please see lines 251-261.)

           Supplement Table 1. Number of colonoscopy received before and after baseline compared between cohorts with and without hemorrhoid.

Hemorrhoids

Variable

No

N=36864

Yes

N=36864

Colonoscopy

N

Rate

n

Rate

p-value

Before

<0.001

0

35993

97.6

32110

87.1

1

732

1.99

3926

10.7

2

102

0.28

614

1.67

≧3

37

0.10

214

0.58

After

<0.001

0

35115

95.3

26338

71.5

1

1391

3.77

6994

19.0

2

253

0.69

2193

5.95

≧3

105

0.28

1339

3.63

Supplement Table 2. Incidence and hazard ratio of colorectal cancer associated with surgery of hemorrhoids

Variables

N

Event

Rate#

Crude

HR (95% CI)

Adjusted

HR (95% CI) 

  Non-hemorrhoids

36864

131

0.53

1.00

1.00

Hemorrhoids

Surgery

No

27783

280

1.46

2.76(2.24, 3.39)***

2.31(1.88, 2.85)***

Yes

9081

50

0.74

1.38(0.99, 1.91)

1.65(1.19, 2.29)**

                    Rate#, incidence per 1000 person-years; HR, hazard ratio. Adjusted HR: controlling for sex and age.

*P < 0.05, **P < 0.01, ***P < 0.001

Supplement Table 3. Incidence and hazard ratios of colorectal cancer associated with polyp or hemorrhage

 compared between the cohorts with and without hemorrhoids.

Hemorrhoids

No

Yes

Variable

Event

PY

Rate#

Event

PY

Rate#

Crude HR

Adjusted HR†          (95% CI)

(95% CI)

Polyp or hemorrhage of rectum and anus

No

124

245766

0.50

266

228979

1.16

2.29 (1.85, 2.84)***

2.09 (1.69, 2.59)***

Yes

7

1657

4.22

64

29914

2.14

0.50 (0.23, 1.10)

0.59 (0.27, 1.31)

Hemorrhage of gastrointestinal tract, unspecified

No

102

234739

0.43

226

219621

1.03

2.35 (1.86, 2.97)***

2.03 (1.60, 2.59)***

Yes

29

12684

2.29

104

39273

2.65

1.16 (0.77, 1.74)

1.27 (0.83, 1.94)

               Rate#, incidence rate per 1000 person-years; PY person-years, HR hazard ratio. Adjusted HR: adjusted for age,

               sex and comorbidity. *P < 0.05, **P < 0.01, ***P < 0.001.

Reviewer 3 Report

hi, thanks for asking me to review this study, I have no major comments, my main concern was that the authors exclude the immediate post-diagnosis (with haemorrhoids) period, which the authors do (by excluding the first follow-up year). Another concern is potential over-diagnosis of haemorrhoids among patients who are screening or under surveillance with colonoscopy follow-up: the authors acknowledge this but perhaps more can be done in method to reflect how haemorrhoids were defined. is it possible to conduct any form of supplementary analysis, repeating the analysis only for the stratum of patients _without evidence of colonoscopy during follow-up, and the one _with; also in the same vein, is there any evidence of a group of patients with surgically/endoscopically treated vs non-treated haemorrhoids, as if the hypothesis of particle contribution of over-detection is true one would expect these two strata to have different risk (i.e. lower among those with surgically/endoscopically treated haemorrhoids as that group is likely to have presented symptomatically). Bar from these two-three issues it all good to go.

Author Response

#Response to Reviewer 3 Comments

Point 1: My main concern was that the authors exclude the immediate post-diagnosis (with haemorrhoids) period, which the authors do (by excluding the first follow-up year).

 Response 1: Thank you for the comment.

Some colorectal cancer cases appear during immortal period, immediately post-diagnosis with haemorrhoids, might not be associated with haemorrhoid. The cancer incidence might be higher if the immortal period is included. The purpose of excluding the first follow-up year in our study design is to avoid the immortal time bias. In the revision, we have specified:

“Individuals with a history of CRC at baseline or within 1 year of the index date were excluded from both cohorts to avoid the immortal time bias.” (Please see lines 81-83.)

Point 2: Another concern is potential over-diagnosis of haemorrhoids among patients who are screening or under surveillance with colonoscopy follow-up: the authors acknowledge this but perhaps more can be done in method to reflect how haemorrhoids were defined.

Response 2: Thank you for the inspirational comment.

In general, the diagnosis of hemorrhoids in Taiwan is not initiated with colonoscopy. Patients with hemorrhoid complaints are usually diagnosed by clinical inspections, digital examinations, and anoscopes in outpatient visits. Physicians order interventions with medicine or surgery to patients based on the severity of symptoms.

It is true people who underwent colonoscopy would be more likely included in the group with hemorrhoids than in the group without hemorrhoids. We have re-designed our study for cohorts with and without hemorrhoid after performing propensity score matched analyses. Based on these cohorts, we further analyzed the data considering the history of colonoscopy. Please see our findings in the reply to comment point 3 in next page.  

Point 3: Is it possible to conduct any form of supplementary analysis, repeating the analysis only for the stratum of patients _without evidence of colonoscopy during follow-up, and the one _with; also in the same vein, is there any evidence of a group of patients with surgically/endoscopically treated vs non-treated haemorrhoids, as if the hypothesis of particle contribution of over-detection is true one would expect these two strata to have different risk (i.e. lower among those with surgically/endoscopically treated haemorrhoids as that group is likely to have presented symptomatically).

Response 3: Thank you for the inspirational suggestions.

For this revision, we conducted a further analysis comparing colonoscopy exams received between the 2 cohorts at baseline and during the follow-up period. Results show that the haemorrhoid cohort received more colonoscopy exams than comparisons, particularly during the follow-up period.

We also conducted the analysis to evaluate the association with surgery for haemorrhoid. Results in Supplement Table 2 show that the               surgery of haemorrhoid could reduce 49.3% of colorectal cancer incidence, from 1.46 to 0.74 per 1000 person-years. In the revision, we have included the findings in the Discussion section to address the study limitation:

“Fourth, patients with hemorrhoids might be more likely to be screened for cancer and were more likely to be diagnosed with the disease. We have conducted further data analysis to compare the history of colonoscopy examinations between cohorts with without hemorrhoid. Results showed that the hemorrhoid group had used colonoscopy more frequently than comparisons had, particularly during the follow-up period (Supplement Table 1). However, we found that hemorrhoid surgery could reduce 49.3% of colorectal cancer incidence, from 1.46 to 0.74 per 1000 person-years (Supplement Table 2). We also found that the hemorrhoids cohort without hemorrhage of rectum and anus were at higher risk for colorectal cancer with an adjusted HR of 2.09 (Supplement Table 3). These data may imply that hemorrhoid could be associated with a 2-fold increased risk without the surgery. Therefore, hemorrhoids are associated with increased risk of CRC remain reliable.” (Please see lines 251-261.)

           Supplement Table 1. Number of colonoscopy received before and after baseline

        compared between cohorts with and without hemorrhoid.

Hemorrhoids

Variable

No

N=36864

Yes

N=36864

Colonoscopy

N

Rate

n

Rate

p-value

Before

<0.001

0

35993

97.6

32110

87.1

1

732

1.99

3926

10.7

2

102

0.28

614

1.67

≧3

37

0.10

214

0.58

After

<0.001

0

35115

95.3

26338

71.5

1

1391

3.77

6994

19.0

2

253

0.69

2193

5.95

≧3

105

0.28

1339

3.63

Supplement Table 2. Incidence and hazard ratio of colorectal cancer associated with surgery of hemorrhoids

Variables

N

Event

Rate#

Crude

HR (95% CI)

Adjusted

HR (95% CI) 

  Non-hemorrhoids

36864

131

0.53

1.00

1.00

Hemorrhoids

Surgery

No

27783

280

1.46

2.76(2.24, 3.39)***

2.31(1.88, 2.85)***

Yes

9081

50

0.74

1.38(0.99, 1.91)

1.65(1.19, 2.29)**

                    Rate#, incidence per 1000 person-years; HR, hazard ratio. Adjusted HR: controlling for sex and age.

                           *P < 0.05, **P < 0.01, ***P < 0.001

Supplement Table 3. Incidence and hazard ratios of colorectal cancer associated with polyp or hemorrhage

 compared between the cohorts with and without hemorrhoids.

Hemorrhoids

No

Yes

Variable

Event

PY

Rate#

Event

PY

Rate#

Crude HR

Adjusted HR†          (95% CI)

(95% CI)

Polyp or hemorrhage of rectum and anus

No

124

245766

0.50

266

228979

1.16

2.29 (1.85, 2.84)***

2.09 (1.69, 2.59)***

Yes

7

1657

4.22

64

29914

2.14

0.50 (0.23, 1.10)

0.59 (0.27, 1.31)

Hemorrhage of gastrointestinal tract, unspecified

No

102

234739

0.43

226

219621

1.03

2.35 (1.86, 2.97)***

2.03 (1.60, 2.59)***

Yes

29

12684

2.29

104

39273

2.65

1.16 (0.77, 1.74)

1.27 (0.83, 1.94)

               Rate#, incidence rate per 1000 person-years; PY person-years, HR hazard ratio. Adjusted HR: adjusted for age,

               sex and comorbidity. *P < 0.05, **P < 0.01, ***P < 0.001.

.                   By the way, an earlier case-control study in the US also found that patients with surgical removal  

         of hemorrhoid are less likely to have anal cancer.

(Tseng, H.F.; Morgenstern, H.; Mack, T.M.; Peters, R.K. Risk factors for anal cancer: results of a population-based case--control study. Cancer Causes Control 2003, 14, 837-846, doi:10.1023/b:caco.0000003837.10664.7f.)

Reviewer 4 Report

Well described in "Limitations" paragraph strenghts and limitations of the study design. Language is easy for readers. Maybe is better to explain the Tseng and Ungerbäck papers on supposed fisiopathologic link between hemorroids and CRC.

Probably the authors should better indicate that molecular biology or epigenetic studies on CRC in patients with hemorrhoids could elucidate a possible pathophysiological link.

Overall, the paper appears well written, the concepts are well explained and easily understood and the tables and figures are clear and sufficient to accompany the work.

Author Response

#Response to Reviewer 4 Comments

Point 1: Maybe is better to explain the Tseng and Ungerbäck papers on supposed physiopathology link between hemorrhoids and colorectal cancer. Probably the authors should better indicate that molecular biology or epigenetic studies on colorectal cancer in patients with hemorrhoids could elucidate a possible pathophysiological link.

Response 1: Thank you for the inspirational suggestion. We have cited these 2 studies and addressed your suggestions in Discussion sections in the revised manuscript.

 “In an earlier case-control study in the US, Tseng et al. reported that hemorrhoids, sexual transmitted disease, physical inactivity, multiple sexual partners are factors associated with an increased the risk of anal cancer [30]. Frequent receptive anal intercourse is likely the physio-pathological evidence leading to inflammation and cancer development. Information on sexual activity and physical inactivity are not available in our data to evaluated these relationships. Another case-control study in San Francisco also found homosexual activity and hemorrhoid are associated with the development of anal and rectal squamous cell carcinoma [31]. Long-lasting hemorrhoids may act similarly to chronic inflammatory dermatoses, which may result in regional inflammatory reactions and evolve into localized squamous-cell carcinomas. In our study, we also found that hemorrhoids significantly increase the risk of distal colon cancer, including sigmoid colon, rectum, anus, and other sites (Table 3). This result revealed that localized inflammation may led to direct tumor formation In addition, epigenetic factors may also associate with the CRC risk. In a Swedish case-control study, Ungerbäck et al. examined polymorphisms in nuclear factor-kappa B (NF?B) pathway-associated genes, the major regulator of inflammation [32]. Results showed that advanced CRC is associated with heterozygous and polymorphic TNFAIP3 (rs6920220), heterozygous NLRP3 (Q705K) and polymorphic NFκB-94 ins/del ATTG genotypes. The proinflammatory NF?B pathway degenerates the tissue, increases the risk of sustained cell and DNA damage and promotes another tumor biomarkers activated in the primary and metastatic CRC. These studies demonstrated that chronic inflammation is associated with the formation of CRC.”

(Please see Discussion section lines 195-215.)

Round 2

Reviewer 2 Report

I would like to thank the authors to have revised the manuscript according to the comments. However, there remains a problem.

1. When performing PS matching, it is better to show the data of the entire cohort before matching, so that the characteristics of the target can be understood. Also, since the large number of the cases, it detects minute differences with p-values, so it might be better to use Standardized Difference. Please consult with your statistician.

Also, please specify in the METHOD the adjusted factors, calipers, and other settings used to the PS matching.

2. Thank you for the additional discussion about colonoscopy. In the supplement data presented, the percentage of patients with hemorrhoids who underwent colonoscopy was about five times higher and more frequent than those without hemorrhoids, which suggests that patients with hemorrhoids have five times more chances to detect colorectal cancer endoscopically. If the results of the comparison were based on the well-adjusted background, including whether or not the patients had undergone colonoscopy, and if colorectal cancer was more common in the group with hemorrhoids, then the authors' argument would make sense. However, the PS matching did not adjust for the presence or absence of colonoscopy. In order to examine the relationship between hemorrhoids and the incidence of colorectal cancer by matching the backgrounds that may affect colorectal cancer diagnosis, please add the colonoscopy (presence/absence) to the matching factors.
